# Occupational exposure to dust and respiratory symptoms among Ethiopian factory workers: A systematic review and meta-analysis

Zemachu Ashuro[1]*, Habtamu Endashaw Hareru[1], Negasa Eshete Soboksa[1], Samson Wakuma Abaya[2], Yifokire Tefera Zele[2]

1 School of Public Health, College of Health Science and Medicine, Dilla University, Dilla, Ethiopia,
2 Department of Preventive Medicine, School of Public Health, College of Health Sciences, Addis Ababa University, Addis Ababa, Ethiopia

* zemash65@gmail.com

## Abstract

### Background

Occupational respiratory disorders are a major global public health concern among workers exposed to dust particles in dust-generating workplaces. Despite fragmented research findings on the magnitude of respiratory problems and the lack of a national occupational respiratory disease recording and reporting system at the Ethiopian factory, the prevalence of respiratory symptoms among factory workers were unknown. Therefore, the aim of this meta-analysis was to summarize and pool estimates from studies that reported the prevalence of respiratory symptoms and predictors among Ethiopian factory workers who worked in dusty environments.

### Methods

A systematic literature searches were conducted using electronic databases (PubMed, Science Direct, African Journals Online, and Web of Science). The primary and secondary outcomes were prevalence of respiratory symptoms and predictors, respectively. The STATA version 17 was used to analyze the data. A random effect meta-analysis model was used. Eggers test with p-value less than 5%, as well as the funnel plot, were used to assess publication bias.

### Results

The searches yielded 1596 articles, 15 of which were included in the systematic review and meta-analysis. The pooled prevalence of respiratory symptoms among Ethiopian factory workers was 54.96% [95% confidence interval (CI):49.33–60.59%]. Lack of occupational health and safety (OSH) training [Odds Ratio (OR) = 2.34, 95%CI:1.56–3.52], work experience of over 5 years [OR = 3.19, 95%CI: 1.33–7.65], not using personal protective

**Data Availability Statement:** All relevant data are within the paper and its Supporting information files.

**Funding:** The authors received no specific funding for this work.

**Competing interests:** The authors have declared that no competing interests exist.

**Abbreviations:** AOR, Adjusted odds ratio; CI, Confidence interval; GDP, Gross domestic product; NOQS, Newcastle Ottawa Quality Score; OSH, Occupational Safety and Health; PPE, Personal Protective Equipment; PRISMA, Preferred Reporting Items for Systematic Reviews and Meta-Analyses; SNNPR, Southern Nations, Nationalities and Peoples Region.

equipment (PPE) [OR = 1.76, 95%CI:1.30–2.39], and working more than eight hours per day [OR = 1.89, 95%CI:1.16–3.05] were all significant predictors of respiratory symptoms.

## Conclusion

The prevalence of respiratory symptom was found to be high in Ethiopian factory workers. To prevent workers from being exposed to dust, regular provision and monitoring of PPE use, workers OSH training, and adequate ventilation in the workplace should be implemented.

## Introduction

According to a 2017 global estimate, 2.78 million workers die each year as a result of occupational accidents and diseases, with 2.4 million of these deaths being disease-related. It is estimated that lost work days account for nearly 4% of global Gross domestic product (GDP), with some countries accounting for 6% or more [1]. Occupational respiratory disorders are a serious global public health problem, accounting for up to 30% of all documented work-related deaths and having a 50% prevalence among employees in high-risk sectors such as mining, construction, and dust-generating works [2].

Dust is defined as small, dry, solid particles that are emitted into the atmosphere by natural forces such as wind and volcanic eruptions, as well as mechanical or man-made processes such as crushing, grinding, milling, drilling, demolition, shoveling, conveying, screening, bagging, and sweeping. Dust particles typically range in size from 1 to 100 micrometer in diameter and settle slowly under the influence of gravity [3].

Organic dusts are dusts derived from microorganisms (bacteria, fungi, viruses, and protozoa) and their metabolites (mycotoxins, peptidoglycans, endotoxins, glucans, enzymes, and so on), animal origin, and plant (vegetable) origin (flour, wood, cotton and tea dusts, pollens vegetable fibers, and epidermis) [4, 5]. The presence of pollutants (bacteria, molds, dust, and pollen) in the workplace pollutes the indoor air quality. Dust exposure in the workplace can have a negative impact on the respiratory health of industrial workers as well as the productivity of industry sectors [6].

A systematic review conducted in Ethiopia revealed that workers' exposure to dust in textile and cement factories far exceeded the international permissible limits [7]. Inhaling dust causes inflammatory reactions in the respiratory system [8, 9]. In developed, low- and middle-income countries, where industry sectors are expanding, respiratory health problems are becoming more common, as are poor occupational health and safety practices by both employees and employers [10].

Lack of trained occupational health professionals, insufficient or nonexistent health legislation and basic services, as well as weak industrial infrastructure and controls, were the major challenges for occupational respiratory disease surveillance, reporting, and recording in low- and middle-income countries [11]. The magnitude of reported respiratory symptoms varies from country to country and factory to factory. The prevalence of occupational respiratory symptoms among cement factory workers in India was 54.4% [12] and 21.1% in Udayapur cement factory workers in Eastern Nepal [13]. According to findings from flour mill factories, the prevalence of respiratory symptoms among factory workers 22% in the United Kingdom [14], 28% in Iran [15], 45% in Nigeria [16] and 90% in Egypt [17]. The prevalence of respiratory symptoms among textile factory workers was 62% in Nigeria [18], 59% in Egypt [19], 53% in Bangladesh [20], and 26% in Iran [21].

Another study of wood workers found that the prevalence of respiratory symptoms was 29.9% in the North East of Thailand [22] and 68% in South-South Nigeria [23]. Furthermore, 61.54% of paper factory workers in Sweden experienced respiratory symptoms [24]. In Ethiopia, the lowest prevalence of respiratory symptoms was reported among textile factory workers (36.8%) [25], while the highest was reported among wood factory workers (69.8%) [26]. However, due to the fragmented findings of primary studies and the lack of a national occupational disease recording and reporting system, estimating the magnitude of respiratory symptoms and risk factors among factory workers at the national level is difficult. Therefore, the objective of this systematic review and meta-analysis was to estimate the national pooled prevalence of respiratory symptoms and risk factors among Ethiopian factory workers. This systematic review and meta-analysis findings will help in the development of appropriate occupational safety and health policies and programs to prevent occupational respiratory disease by implementing appropriate dust exposure prevention and control measures in factories.

## Methods

### Reporting and protocol registration

This systematic review and meta-analysis, which is available at: https://www.crd.york.ac.uk/prospero/displayrecord.php?ID=CRD42022363183, was conducted in accordance with the PRISMA (Preferred Reporting Items for Systematic Reviews and Meta-Analysis statement) guidelines to estimate the pooled prevalence of respiratory symptoms and associated factors among factory or industry workers [27, 28].

### Literature search strategy

We conducted a systematic search of various electronic databases, including PubMed, Science Direct, African Journals Online and Web of science, to estimate the pooled prevalence of respiratory symptoms and associated factors among Ethiopian factory workers. In addition, we searched the Ethiopian University electronic library for unpublished studies. The search was restricted to human studies conducted between 2014 and 2022, as well as full English version articles. We used Boolean operators "AND" and "OR.". The search was conducted independently by two authors (ZA and HEH) using the (MeSH) terms and Text Word (S1 Table). We also looked up "grey literature" on Google Scholar (references not found in PubMed, Science Direct, African Journals Online, or Web of Science). In addition, we searched Ethiopian university databases for unpublished studies. Furthermore, when an article lacked sufficient data, corresponding authors of the research article were contacted via email.

### Eligibility criteria

**Inclusion criteria.** *Study settings*. Only studies conducted among Ethiopian factory workers.

*Publication condition*. Both published and unpublished articles were included.

*Study design*. All observational study designs (cross-sectional, case control and cohort) reporting the prevalence of the respiratory symptoms and its associated factors were eligible for this systematic review and meta-analysis.

*Language*. Only studies published in the English language were considered.

*Population*. Participants /workers/ whose age was 18 and greater.

*Outcome*. Studies reported the odds of respiratory symptoms and associated factors related to dust exposure with corresponding 95% confidence interval.

*Publication year.* Articles that were conducted between 2014 and 2022 were included in this systematic review and meta-analysis to generate more recent information that will be helpful for policymakers.

**Exclusion criteria.** Studies that did not show clear data regarding the respiratory symptoms, abstract without full-text, qualitative studies, editorials, and commentaries were excluded from this systematic review and meta-analysis.

## Data screening and extraction

The studies that met the inclusion criteria were reviewed. To evaluate the identified studies, we used a two-level screening approach. To begin, titles and abstracts were reviewed for eligibility. This was done independently by ZA and HEH. Second, the full articles were assessed. Before starting the data extraction, the extraction format was prepared in Microsoft Excel Spreadsheet. The author's name, year of publication, type of factory, region, study design, sample size, response rate, and prevalence of respiratory symptoms for primary outcome (magnitude of respiratory symptoms) were all included in the data extraction format. A two-by-two table was used to extract data from included studies for the second outcome (predictors of the respiratory symptoms). Studies that met the inclusion criteria were screened and extracted by ZA and HEH using a standardized data extraction format, and any disagreements were resolved through discussion with the other investigators (NES, SWA and YTZ).

## Outcome measurement

Respiratory symptoms are defined as experiencing one or more of the following symptoms as a result of occupational exposure: cough, phlegm, wheezing, dyspnea, bronchitis, shortness of breath, and chest pain [29–31]. The pooled prevalence of respiratory symptoms was calculated by dividing the total number of workers with respiratory symptoms by the total number of workers in the study and multiplying the result by 100. The factors associated with respiratory symptoms were measured using the odds ratio. We used two-by-two tables to calculate the odds ratio from primary studies.

## Quality assessment

The Newcastle-Ottawa quality assessment scale was used to evaluate the quality of the included primary studies. This tool has three domains: selection (a maximum of 5 stars), comparability (a maximum of 2 stars), and outcome (a maximum of 3 stars). The studies score ranges from 0–10 for each study. A study can receive a maximum of 10 stars, indicating the highest level of quality [32]. All included studies were evaluated independently by the authors.

**Statistical analysis.** To analyze the retrieved data, Stata Corporation, College Station, TX: StataCorp LLC, software version 17.0 was used. The $I^2$ test was used to determine the heterogeneity within the primary studies that were included. We used a random effect model to estimate the pooled prevalence of respiratory symptoms among factory workers because there was significant heterogeneity among included studies. A subgroup analysis was performed by region and factory type to reduce the random variations between the primary study's point estimates. Sensitivity analysis was performed to determine the effect of a single study on the pooled estimate of outcome. Eggers test with p-value less than 5%, as well as the funnel plot, were used to assess publication bias. The pooled point prevalence with 95%Cl was presented using a forest plot. A meta-regression analysis was carried out to identify potential contributors to the between-study heterogeneity. Univariable analysis was performed on variables such as sample size, factory type, and study settings (region). We used p-values less than 0.25 in univariable meta-regression analysis to declare statistically significantly associated variables.

## Results

### Search results

We found 1596 articles in the databases PubMed, Science Direct, Google Scholar, African Journals Online, and Web of Science. However, for various reasons, only 15 articles reporting the prevalence of respiratory symptoms and associated factors among factory workers were included in the final systemic review and meta-analysis procedure (Fig 1).

### Descriptive results of eligible studies

This systematic review and meta-analysis included 15 primary studies conducted in Ethiopia and published between 2014 and 2022, with a total of 4129 study participants. The majority of the included studies were conducted in Addis Abeba (n = 5) [26, 33–36] and Amhara Region (n = 5) [25, 37–40], with two studies conducted in Southern Nations, Nationalities and Peoples Region (SNNPR) [41, 42]. All included studies in this systematic review and meta-analysis were cross-sectional studies conducted among Ethiopian factory/industry workers, with the smallest sample size of 51 reported from a study conducted in SNNPR in a textile factory [41] and the largest sample size of 496 reported from a study conducted in Addis Ababa among wood factory workers [26]. The prevalence of respiratory symptoms reported from the primary studies ranged from 27.7% among flour mill factory workers [34] to 69.8% among wood factory workers [26]. The included studies' quality scores range from 7 to 10 (Table 1).

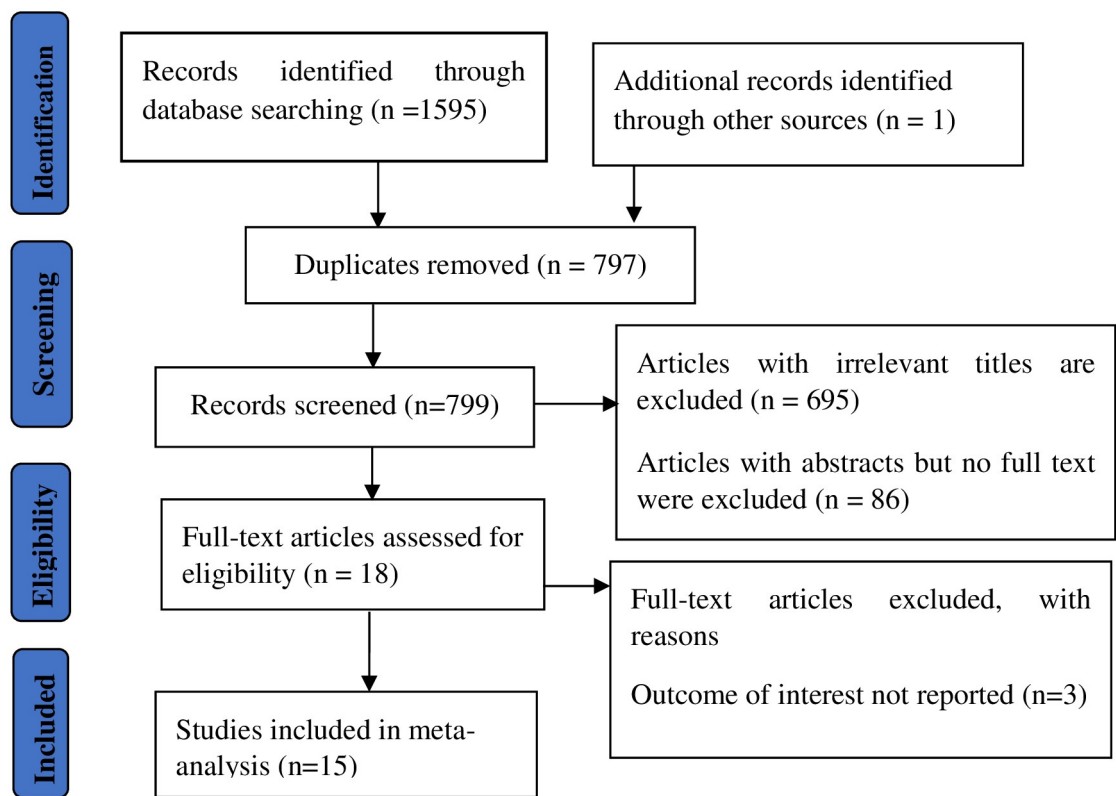

**Fig 1. Flow diagram of articles search and selection criteria.**

**Table 1. Summary characteristics of the included studies to estimate the prevalence and associated factors of respiratory symptoms in Ethiopia, 2014–2022 (n = 15).**

| Author | Year | Region | Sample size | Type of factory | Study design | Response rate (%) | NOQS |
|---|---|---|---|---|---|---|---|
| Alemseged et al. [33] | 2020 | Addis Ababa | 415 | Flour | Cross-sectional | 97.9 | 10 |
| Demeke and Haile [34] | 2018 | Addis Ababa | 54 | Flour | Cross-sectional | 100 | 7 |
| Mekonnen et al. [37] | 2021 | Amhara | 280 | Flour | Cross-sectional | 100 | 9 |
| Lagiso et al. [42] | 2020 | SNNPR | 196 | Flour | Cross-sectional | 93.3 | 8 |
| Kanko et al. [41] | 2017 | SNNPR | 51 | Textile | Cross-sectional | 99.7 | 7 |
| Zele et al. [38] | 2020 | Amhara | 303 | Textile | Cross-sectional | 100 | 10 |
| Daba Wami et al. [39] | 2018 | Amhara | 270 | Textile | Cross-sectional | 99 | 8 |
| Reta Demissie [43] | 2019 | Oromia | 70 | Wood | Cross-sectional | 97.8 | 7 |
| Jabur et al. [36] | 2022 | Addis Abeba | 230 | Wood | Cross-sectional | 100 | 8 |
| Awoke et al. [26] | 2021 | Addis Abeba | 496 | Wood | Cross-sectional | 99.1 | 10 |
| Meskele et al. [35] | Unpublished | Addis Abeba | 206 | Paper | Cross-sectional | 98 | 7 |
| Kifle et al. [25] | 2020 | Amhara | 383 | Cement | Cross-sectional | 91.7 | 9 |
| Gizaw et al. [40] | 2016 | Amhara | 404 | Cement | Cross-sectional | 100 | 10 |
| Siyoum et al. [44] | 2014 | Oromia | 266 | Cement | Cross-sectional | 97.8 | 8 |
| Mekasha et al. [45] | 2018 | Oromia | 309 | Cement | Cross-sectional | 97 | 8 |

NOQS: Newcastle Ottawa Quality Score, SNNP: Southern Nations, Nationalities, and Peoples.

## Meta-analysis

In this meta-analysis, individual study prevalence estimates range from 27.70% to 69.80%, with an overall pooled prevalence of 54.96% (95% CI: 49.33–60.59%; $I^2$ = 92.4%, p<0.000) and individual study weights ranging from 5.59% to 7.19% (Fig 2).

## Subgroup analysis

Subgroup analysis was carried out for both region and factory type. The sub-group analysis revealed that the SNNPR had the highest prevalence of respiratory symptoms, with a prevalence of 58.65% (95% CI: 51.75, 65.57%), and the Amhara region had the lowest prevalence, with a prevalence of 53.07% (95% CI: 42.62, 63.52%) (Fig 3).

The other subgroup analysis was done for the type of factory involved. The highest prevalence of respiratory symptoms was reported among wood factory workers (60.28% (95% CI: 48.04, 72.51%), and the lowest prevalence was reported among textile factory workers (49.78% (95% CI: 39.56, 59.99%) (Fig 4).

**Publication bias and small study effect assessment.** We used Egger's test and a funnel plot to assess publication bias. The analysis revealed a symmetrical funnel plot indicating no publication bias (Fig 5), and Egger's test for small-study effect revealed no small-study effects with a p-value of 0.214 (Table 2). We did not perform trim and fill analysis because there is no publication bias.

## Sensitivity analysis

According to the results of the sensitivity analysis, individual studies had no significant impact on the overall pooled prevalence of respiratory symptoms among factory workers (Fig 6).

## Meta-regression

Meta-regression analysis was performed on the variables included, which were sample size, factory type, and study settings (region). We used p-values less than 0.25 in univariable meta-

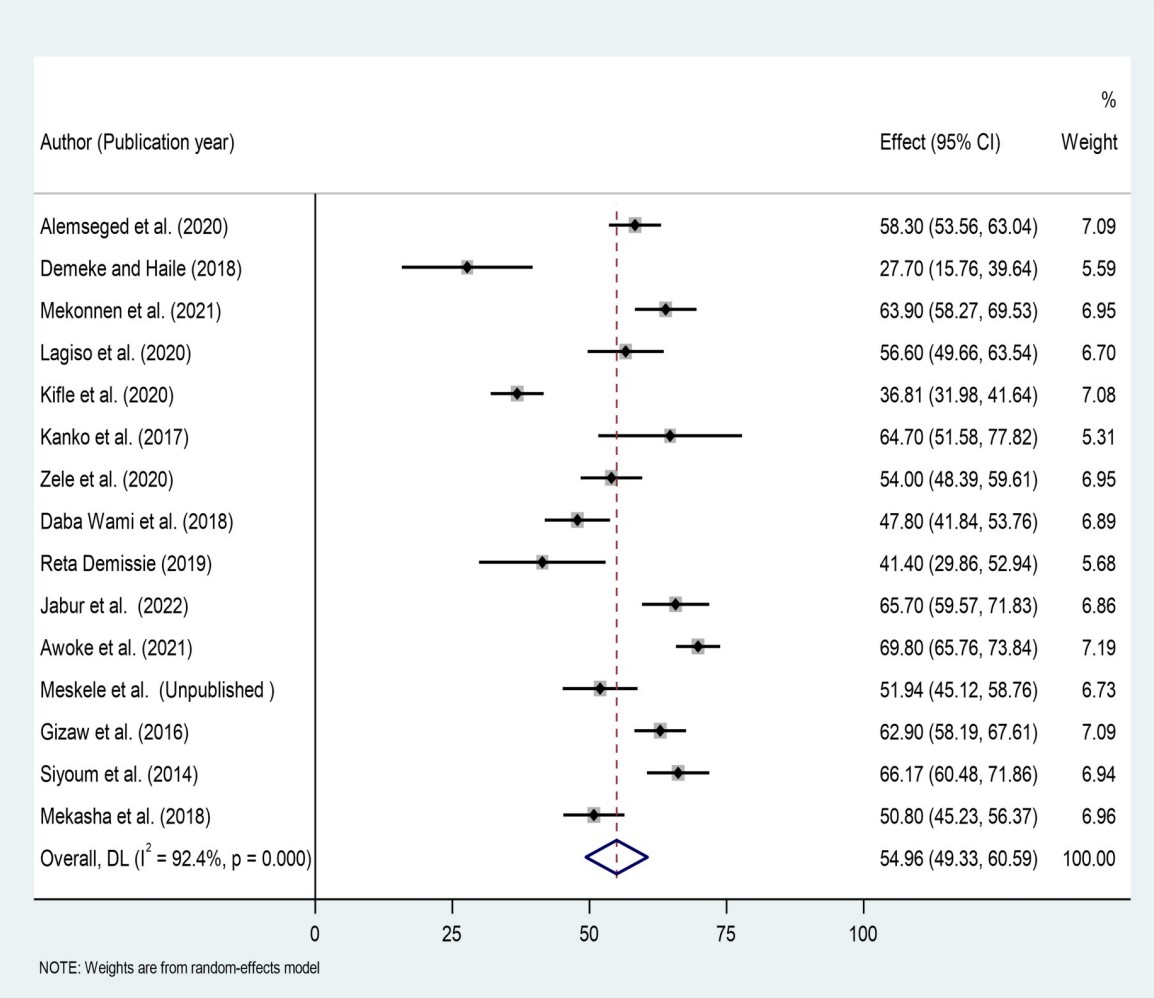

**Fig 2. Estimates of the prevalence of respiratory symptoms in Ethiopian factory workers using a forest plot.**

regression analysis to declare statistically significantly associated variables. However, in this study, all variables were not significantly associated with the prevalence of respiratory symptoms in the univariable meta-regression analysis (Table 3).

## Factors associated with the occurrence of respiratory symptoms

From the primary studies, we identified a number of factors associated with respiratory symptoms. This meta-analysis included variables found to have a significant association with respiratory symptoms in at least four primary studies. In this systematic review and meta-analysis, occupational safety and health (OSH) training (OR = 2.34, 95%CI: 1.56–3.52), work experience (OR = 3.19, 95%CI: 1.33–7.65), PPE use (OR = 1.76, 95%CI: 1.30–2.39), and working hours (OR = 1.89, 95%CI: 1.16–3.05) were significant predictors of respiratory symptoms among Ethiopian factory workers.

**Association between OSH training and respiratory symptoms.** In this meta-analysis, we included seven studies [25, 26, 35, 37, 40, 44] to investigate the relationship between workers' OSH training and respiratory symptoms. According to the findings of these seven studies,

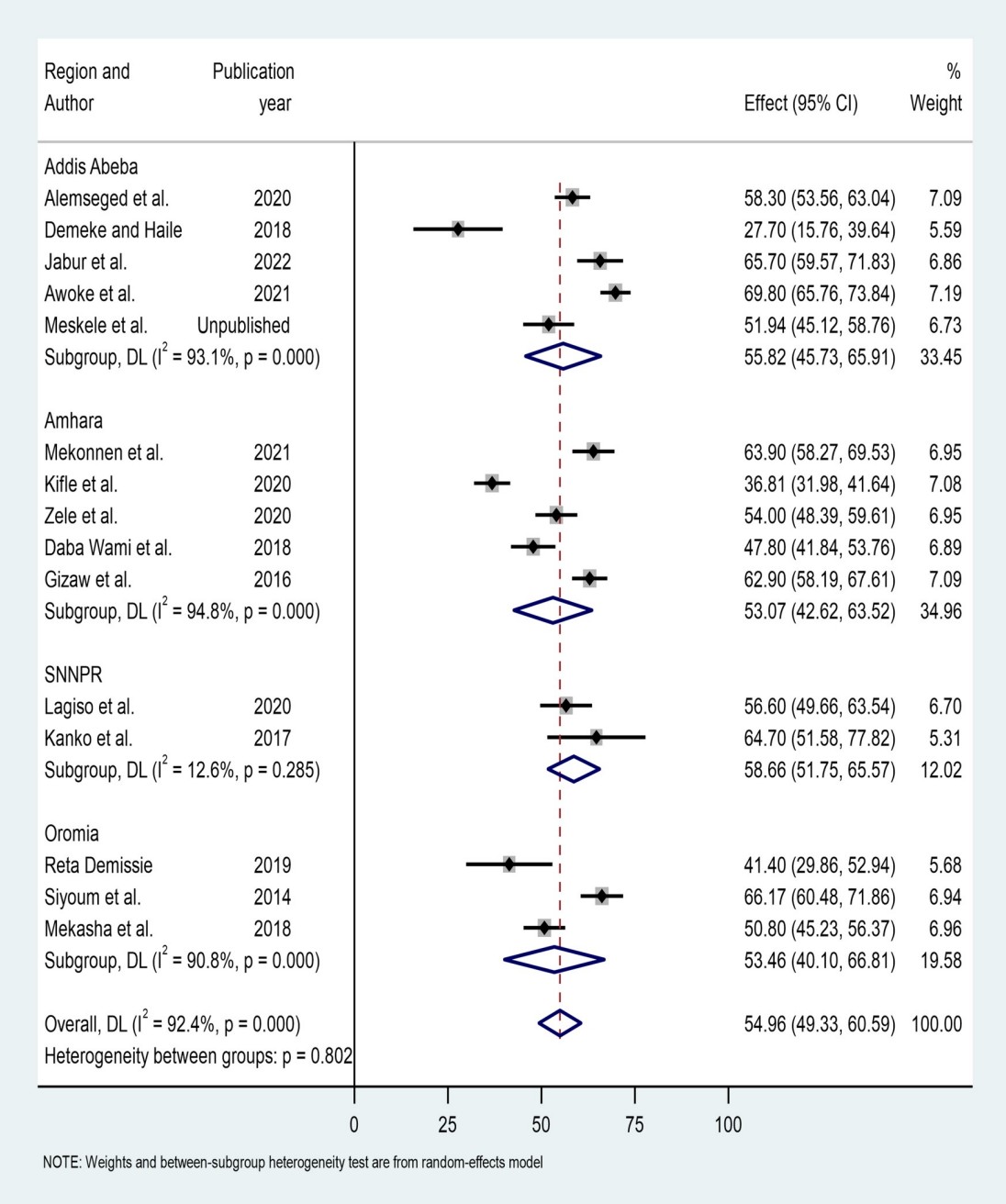

**Fig 3. Forest plot on respiratory symptoms prevalence estimates subgroup analysis by region.**

workers who did not receive OSH training were 2.34 times more likely to develop respiratory symptoms than workers who did receive OSH training [OR = 2.34, 95%CI: 1.56–3.52] (Fig 7).

## Association between work experiences and respiratory symptoms among Ethiopian factory workers

We included six primary studies [26, 36, 37, 39, 40, 42, 44] conducted in Ethiopian factory workers to investigate the association between factory workers' work experiences and the

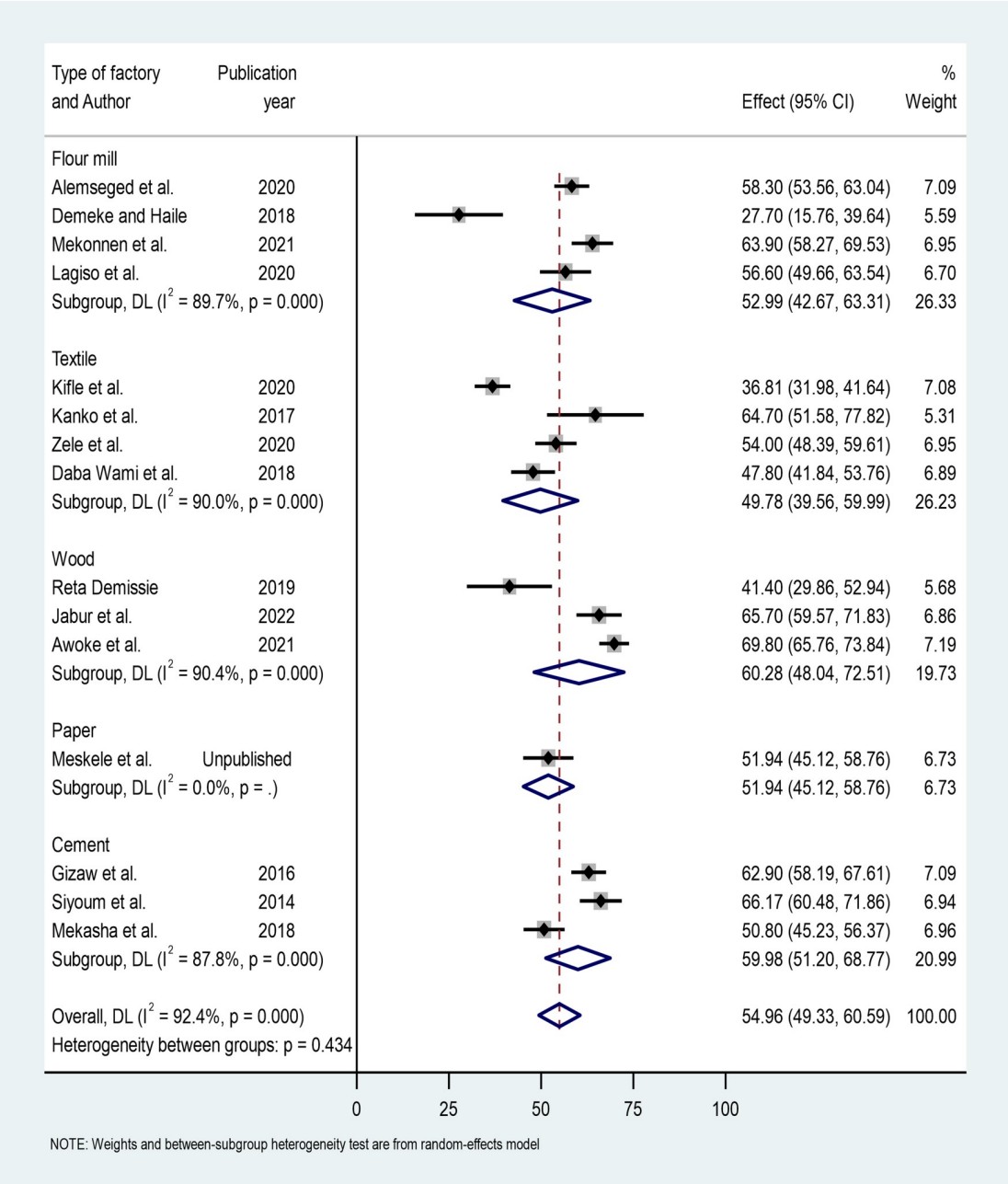

**Fig 4. Subgroup analysis by factory type using a forest plot to estimate the prevalence of respiratory symptoms.**

occurrence of respiratory symptoms. The results of a meta-analysis of six studies revealed that the odds of respiratory symptoms were 3.19 times higher among workers with work experiences greater than five years (>5 years) than among workers with work experiences less than or equal to five years (OR = 3.19, 95%CI: 1.33–7.65) (Fig 8).

**Association between gender and occurrence of respiratory symptoms.** Eight studies were used to determine the association between study participants' gender and the occurrence of respiratory symptoms among factory workers [25, 35–37, 39, 42, 44]. A meta-analysis of the

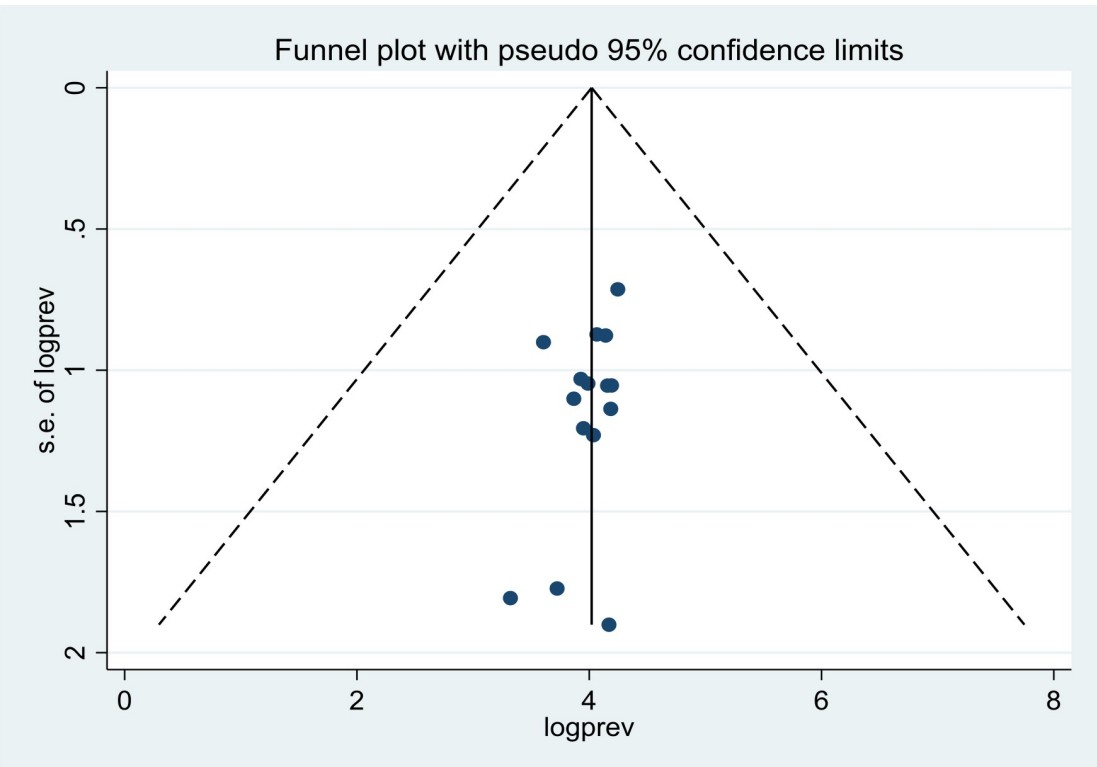

**Fig 5. Funnel plot that assesses publication bias.**

findings of the eight included studies revealed that there was no statistically significant association between gender and respiratory symptoms among Ethiopian factory workers (OR = 1.25, 95%CI: 0.83–1.89) (Fig 9).

## Association between PPE use and occurrence of respiratory symptoms

We investigated five primary studies conducted among factory workers in different Ethiopian factories [25, 33, 35, 39]. The findings of this meta-analysis revealed that workers who did not use PPE in the workplace were 1.76 times more likely to develop respiratory symptoms than those who did (OR = 1.76, 95%CI: 1.30–2.39) (Fig 10).

**Association between working hours and the occurrence of respiratory symptoms.** In this meta-analysis, we included four primary studies conducted in different Ethiopian factories [26, 35, 37, 42, 44]. According to the meta-analysis findings, workers who worked more than 8 hours per day in the factory were 1.89 times more likely to develop respiratory symptoms than those who worked 8 hours or less in the factory (OR = 1.89, 95%CI: 1.16–3.05) (Fig 11).

**Association between biofuel use and respiratory symptoms among Ethiopian factory workers.** Four studies were included in the meta-analysis to investigate the association

**Table 2. Eggers test for publication bias assessment.**

| Standard Effect | Coefficient | t-value | p-value | 95% CI |
|---|---|---|---|---|
| Slope | 69.17 | 7.16 | 0.000 | 48.31–90.03 |
| Bias | -4.22 | -1.31 | 0.214 | -11.20–2.75 |

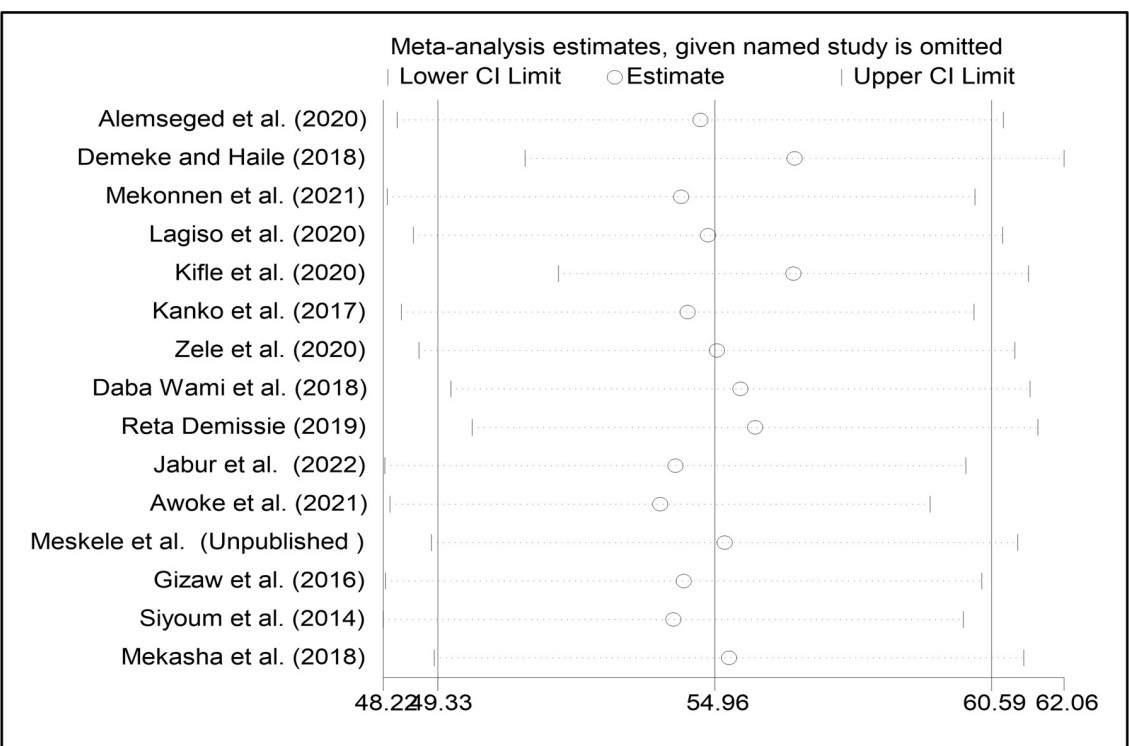

**Fig 6. Sensitivity analysis of included studies.**

**Table 3. Univariable meta-regression model.**

| Variables | Coefficient | P-value | 95% CI |
|---|---|---|---|
| **Type of factory** | | | |
| Cement | 0.08 | 0.843 | -0.555–1.216 |
| Flour mill | 0.33 | 0.464 | -0795-0.938 |
| Textile | 0.07 | 0.872 | -0.964–0767 |
| Wood | 0.36 | 0.425 | -0.530–1.259 |
| Paper | Reference | | |
| **Constant** | **0.08** | **0.843** | **-0.693–0.848** |
| **Study settings (region)** | | | |
| Addis Ababa | -0.16 | 0.698 | -0.959–0.642 |
| Amhara | -0.29 | 0.473 | -1.085–0.503 |
| Oromia | -027 | 0.544 | 1.138–0.600 |
| SNNPR | Reference | | |
| **Constant** | **0.41** | **0.236** | **-0.271–1.099** |
| **Sample size** | | | |
| <262 | Reference | | |
| ≥262 | 0.05 | 0.832 | -0.419–0.520 |
| **Constant** | **0.18** | **0.322** | **-0.175–0.532** |

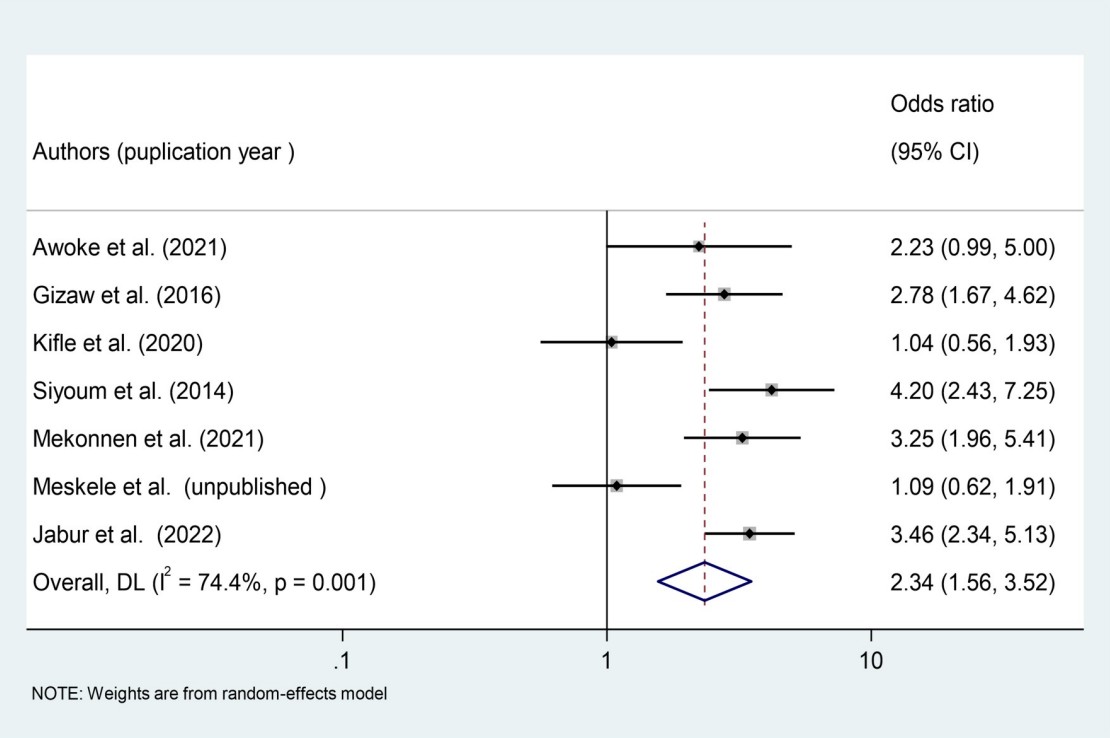

**Fig 7. Association of OSH training with respiratory symptoms among factory workers in Ethiopia, 2014–2022.**

between biofuel use and respiratory symptoms among factory workers [25, 26, 35, 44]. According to the findings of this meta-analysis, the use of bio-fuel for cooking at home was not significantly associated with the occurrence of respiratory symptoms among Ethiopian factory workers (OR = 1.65, 95%CI: 0.72–3.79) (Fig 12).

### Association between educational status and respiratory symptoms among Ethiopian factory workers

We used six primary studies to explore the association between workers' educational status and respiratory symptoms among Ethiopian factory workers [35, 36, 39, 40, 42, 44]. According to a meta-analysis of six primary studies, workers' educational status was not significantly associated with the occurrence of respiratory symptoms among Ethiopian factory workers (OR = 1.18, 95%CI: 0.41–3.37) (Fig 13).

## Discussion

Occupational respiratory disorders are a major global public health concern among factory workers. This systematic review and meta-analysis finding showed that the pooled estimate of respiratory symptoms among factory workers in Ethiopia was 54.96% (95% CI: 49.33–60.59%; $I^2$ = 92.4%, p<0.000). Lack of occupational health and safety training, work experience of over 5 years, not using personal protective equipment, and working more than eight hours per day were all predictors of high prevalence of respiratory symptoms.

The overall estimate of respiratory symptoms among factory workers in Ethiopia is high. The finding of this study was consistent with studies conducted among cement factory workers

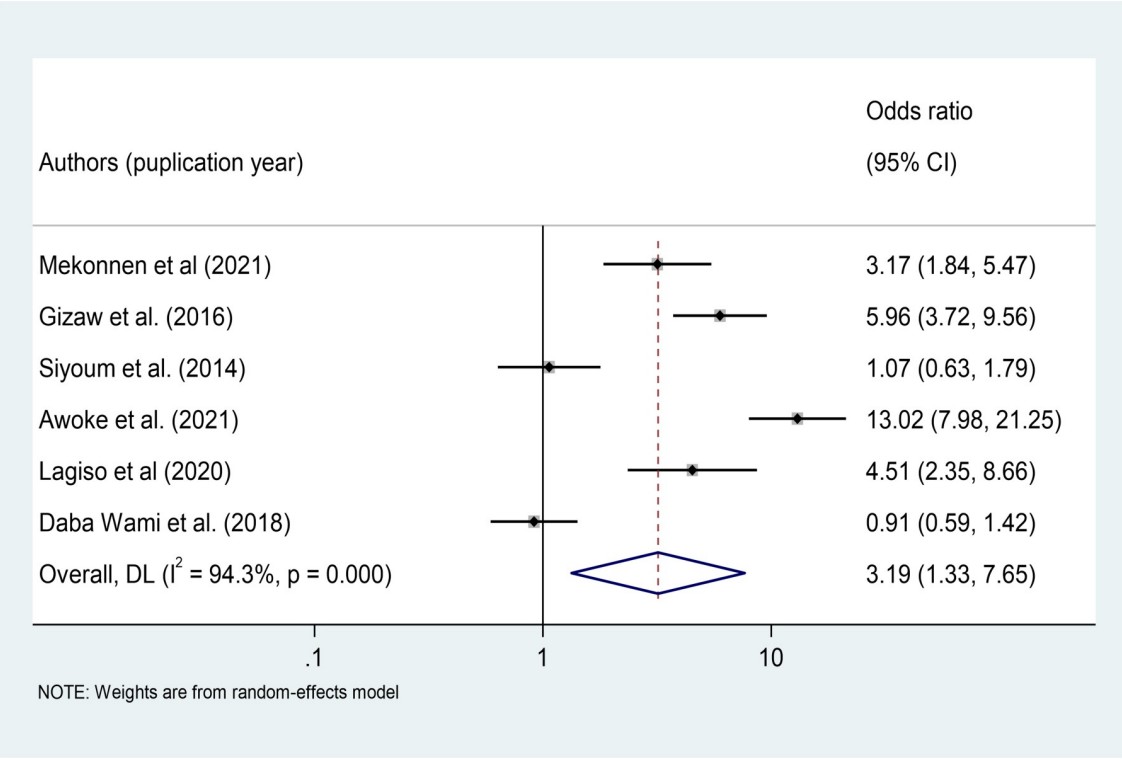

**Fig 8. Association of [40] work experiences with respiratory symptoms among factory workers in Ethiopia, 2014–2022.**

in India (54.4%) [12], textile factory workers in Bangladesh (53%) [20], and wood workers in Cameroon (51%) [46]. However, the finding of this study was higher than the findings of studies conducted among cement factory workers in eastern Nepal (21.1%), flour mill factory workers in the United Kingdom (22%) [14], Iran (28%) [15], and Nigeria (45%) [16], textile factory workers in Iran (26%) [21], and a study conducted among wood workers in Thailand's North East (29.9%) [22], whereas the result of this study was lower than those of studies conducted among Egyptian factory workers in flour mills (90%) [17], Nigerian factory workers in textiles (62%) [18], study conducted among South-South Nigerian wood factory workers (68%) [23], and in Sweden, paper factory (61.54%) [24]. Differences in the work environment, the type of factory, and occupational health and safety practices could all be contributing factors to the discrepancy.

The pooled prevalence of respiratory symptoms was higher among wood factory workers 60.28% (95%CI: 48.04, 72.51%) than among textile factory workers 49.7% (95% CI: 39.56, 59.99%), according to a subgroup analysis of studies by factory type. Workers who did not receive OSH training were 2.34 times more likely to develop respiratory symptoms than workers who did receive OSH training. This study finding was supported by studies conducted in Egypt [47] and Norway [48]. The possible explanation could be that training may provide workers with awareness of how to protect themselves from work-related hazards exposure in the workplace, as well as behavioral change in workers toward occupational safety and health practices.

The result of this study revealed that the odds of respiratory symptoms were 3.19 times higher among workers with work experiences greater than five years (>5 years) than among workers with work experiences less than or equal to five years. This study finding was

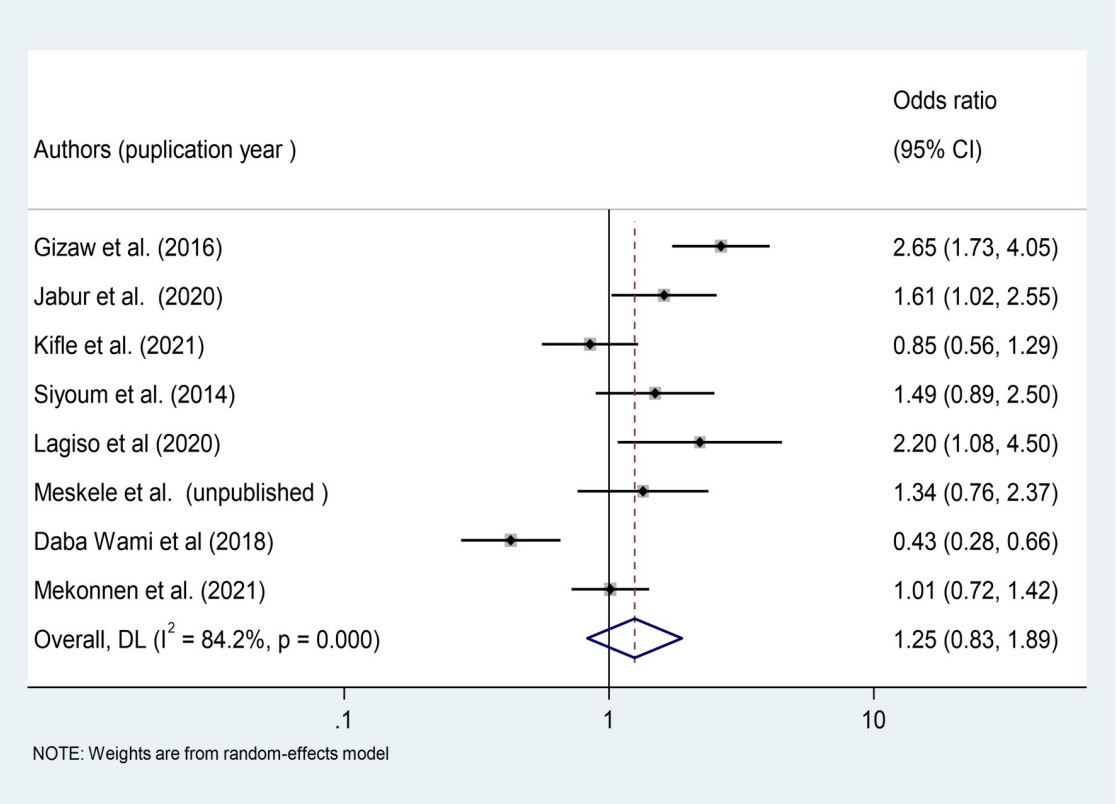

**Fig 9. Association of gender with respiratory symptoms among factory workers in Ethiopia, 2014–2022.**

supported with study conducted in Eastern Nepal [13] and in Egypt [17]. The high prevalence of respiratory symptoms could be attributed to increased dust accumulation in the respiratory system as a result of long-term exposure to dust in the workplace.

According to the systematic review and meta-analysis finding, workers who worked more than 8 hours per day in the dust environment of a factory were 1.89 times more likely to develop respiratory symptoms than those who worked 8 hours or less in the dust environment of a factory. This finding was in line with the study conducted in Thailand, Bangkok among garment workers [49]. The possible explanation is that workers who work in dusty environments for extended periods of time without wearing PPE increase their risk of exposure in the workplace.

According to the finding of this systematic review and meta-analysis, workers who did not use PPE in the workplace were 1.76 times more likely to develop respiratory symptoms than workers who did. This study finding were supported with those of studies conducted in Dhaka, Bangladesh [50], the United Arab Emirates cement factory [51], and the furniture industry workers in Indonesia [52]. The possible explanations for the high prevalence of respiratory symptoms might be workers exposed to different dust particles at work environment due to the lack of PPE in work environment increases workers exposure. This systematic review and meta-analysis is timely and will help to improve occupational safety and health practices, as well as promote and maintain worker health in industry. Because low-income countries have a low level of a national occupational disease recording and reporting system, estimating the magnitude of the problem is difficult. However, this result may help

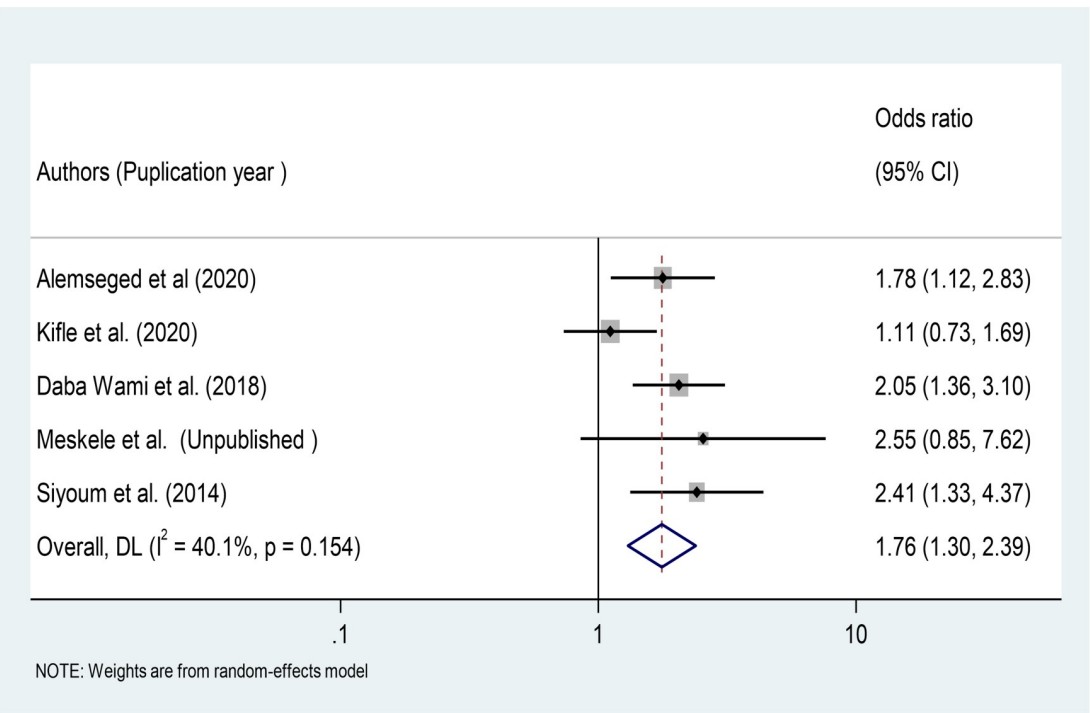

**Fig 10. Association of PPE use with respiratory symptoms among factory workers in Ethiopia, 2014–2022.**

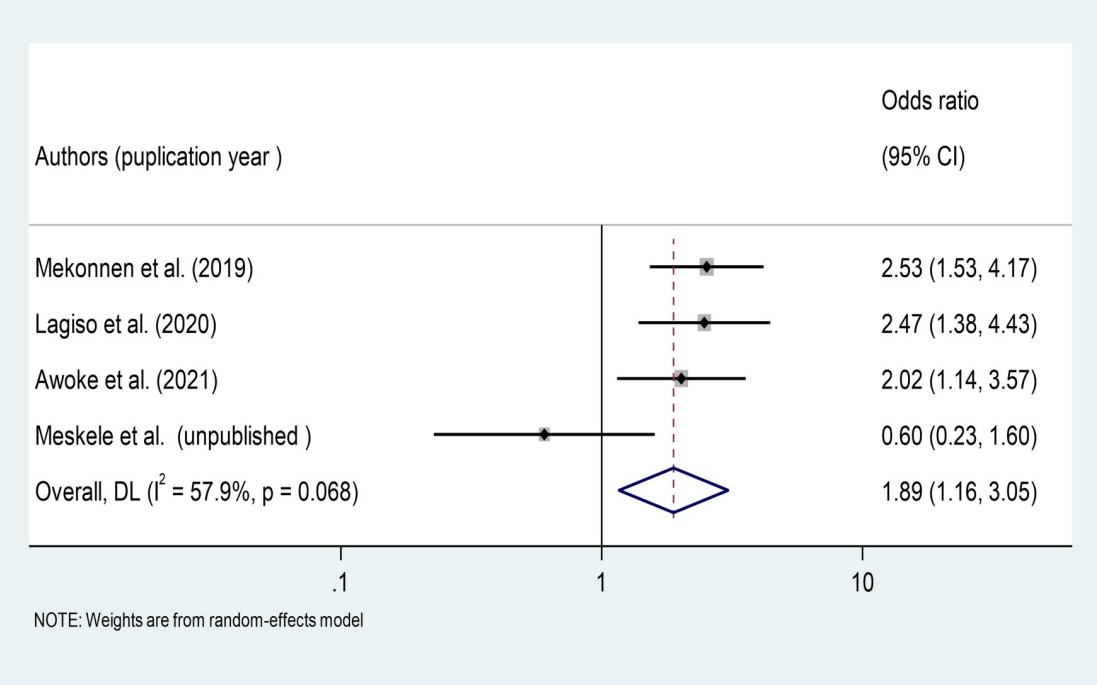

**Fig 11. Association of working hours with respiratory symptoms among factory workers in Ethiopia, 2014–2022.**

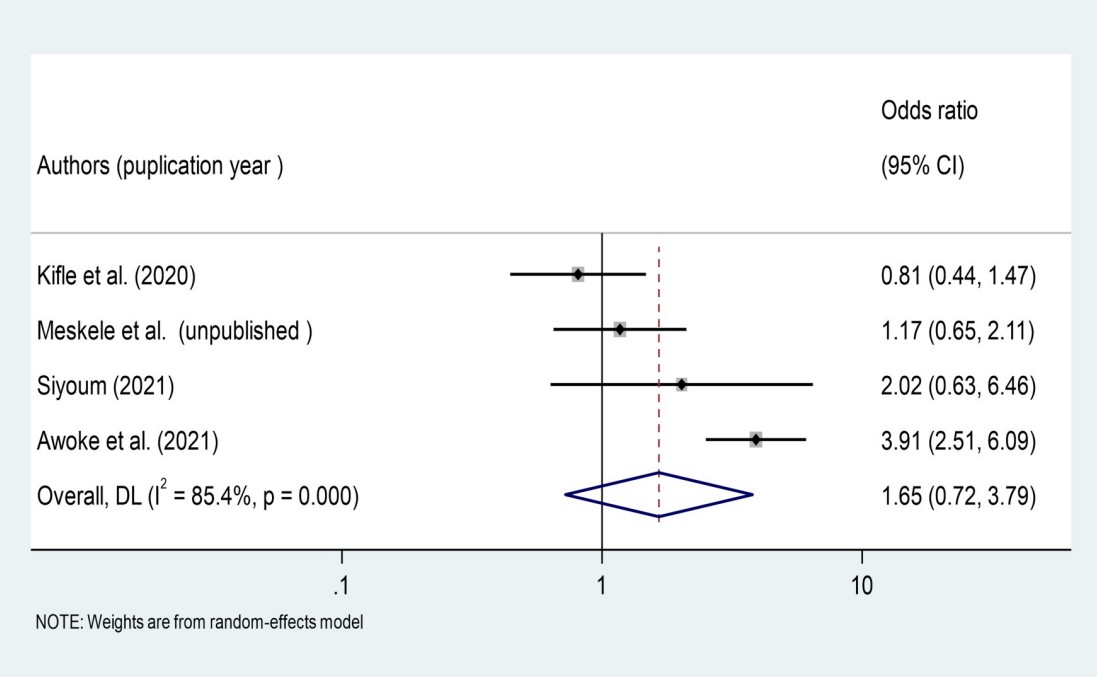

**Fig 12. Association of biofuel used with respiratory symptoms among factory workers in Ethiopia, 2014–2022.**

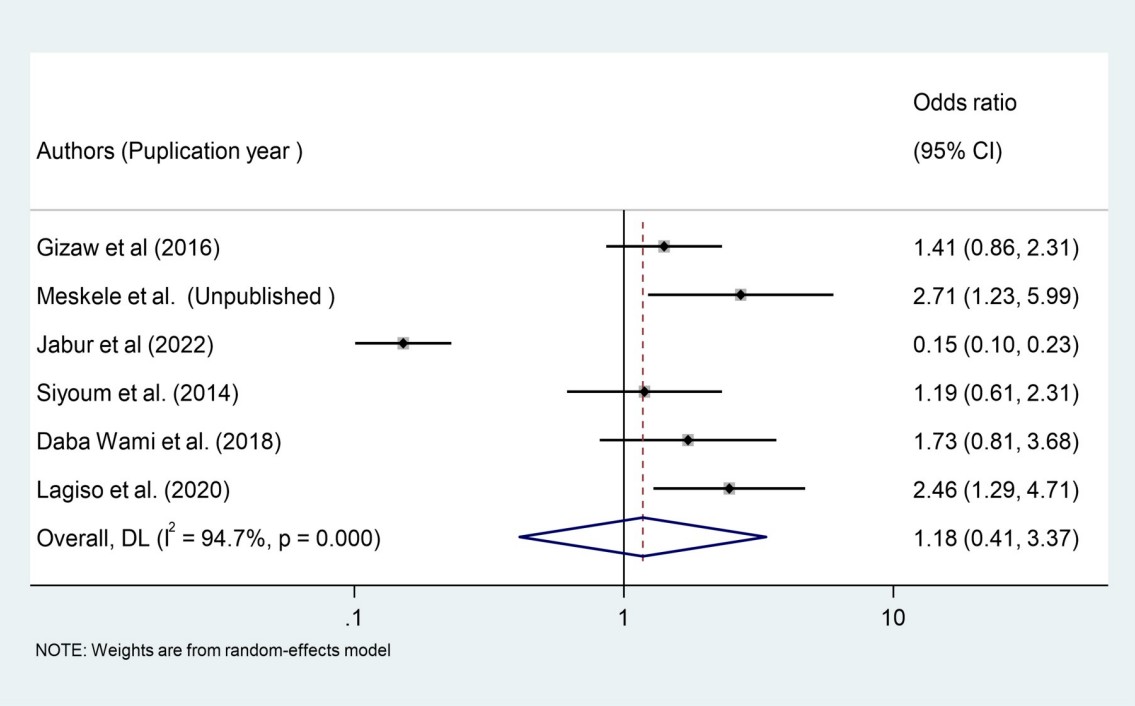

**Fig 13. Association of educational status with respiratory symptoms among factory workers in Ethiopia, 2014–2022.**

policymakers implement appropriate occupational safety and health interventions. The limitations of this systematic review and meta-analysis were that there was a high degree of heterogeneity among the included studies ($I^2$ statistic = 92.4%, p<0.000) and that only studies published in English were included, which may not have traced all studies. Furthermore, the studies included in this systematic review and meta-analysis were from only four regions, which do not adequately represent the rest of the country, and some studies had small sample sizes, which may have an effect on the pooled estimate of respiratory symptoms.

## Conclusions

This meta-analysis and systematic review revealed a high prevalence of respiratory symptoms among factory workers. Lack of occupational health and safety training, work experience of over 5 years, not using personal protective equipment, and working more than eight hours per day were all predictors of high prevalence of respiratory symptoms. Therefore, regular provision and monitoring of personal protective equipment use, provision of occupational safety and health training for workers, and provision of adequate ventilation in the workplace should be implemented to prevent workers' exposure to dust. The included studies in this systematic review and meta-analysis were cross-sectional studies that did not show a causal relationship between dust exposure and respiratory effects. To plan an effective control strategy, studies on factory workers should be conducted using a prospective cohort study design. Furthermore, the Federal Ministry of Labor and Social Affairs of Ethiopia, the Federal Ministry of Health of Ethiopia, and other stakeholders should work together to improve occupational health and safety practices at the factory level.

## Supporting information

**S1 Table. Summary of search results from PubMed, Google Scholar, Science Direct, African Journals Online, and Web of Science databases.**
(DOCX)

**S2 Table. The PRISMA 2020 checklist (Preferred reporting items for systematic review and meta-analysis) is an updated guideline for reporting systematic reviews.**
(DOCX)

## Acknowledgments

We thank the original study's authors and participants for their contributions to this systematic review and meta-analysis.

## Author Contributions

**Conceptualization:** Zemachu Ashuro.

**Data curation:** Zemachu Ashuro, Habtamu Endashaw Hareru, Negasa Eshete Soboksa, Samson Wakuma Abaya.

**Formal analysis:** Zemachu Ashuro, Habtamu Endashaw Hareru, Negasa Eshete Soboksa, Yifokire Tefera Zele.

**Funding acquisition:** Zemachu Ashuro, Habtamu Endashaw Hareru, Yifokire Tefera Zele.

**Investigation:** Zemachu Ashuro, Samson Wakuma Abaya, Yifokire Tefera Zele.

**Methodology:** Zemachu Ashuro, Negasa Eshete Soboksa, Samson Wakuma Abaya, Yifokire Tefera Zele.

**Project administration:** Zemachu Ashuro, Samson Wakuma Abaya.

**Resources:** Zemachu Ashuro, Habtamu Endashaw Hareru, Negasa Eshete Soboksa, Samson Wakuma Abaya, Yifokire Tefera Zele.

**Software:** Zemachu Ashuro, Habtamu Endashaw Hareru, Negasa Eshete Soboksa, Samson Wakuma Abaya.

**Supervision:** Zemachu Ashuro, Samson Wakuma Abaya, Yifokire Tefera Zele.

**Validation:** Zemachu Ashuro, Yifokire Tefera Zele.

**Visualization:** Zemachu Ashuro, Yifokire Tefera Zele.

**Writing – original draft:** Zemachu Ashuro, Habtamu Endashaw Hareru, Negasa Eshete Soboksa, Samson Wakuma Abaya, Yifokire Tefera Zele.

**Writing – review & editing:** Zemachu Ashuro, Habtamu Endashaw Hareru, Negasa Eshete Soboksa, Samson Wakuma Abaya.

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
