## [Decision Letter · Decision Letter 0]

14 Feb 2023

PONE-D-22-30181Occupational exposure to dust and Respiratory symptoms among Ethiopian factory workers: a systematic review and meta-analysisPLOS ONE

Dear Dr. Ashuro,

Thank you for submitting your manuscript to PLOS ONE. After careful consideration, we feel that it has merit but does not fully meet PLOS ONE’s publication criteria as it currently stands. Therefore, we invite you to submit a revised version of the manuscript that addresses the points raised during the review process.

We look forward to receiving your revised manuscript.

Kind regards,

Ghulam Md Ashraf, Ph.D.

Academic Editor

PLOS ONE

Journal Requirements:

Reviewers' comments:

Reviewer's Responses to Questions

**Comments to the Author**

1. Is the manuscript technically sound, and do the data support the conclusions?

Reviewer #1: Yes

Reviewer #2: No

2. Has the statistical analysis been performed appropriately and rigorously? 

Reviewer #1: I Don't Know

Reviewer #2: No

3. Have the authors made all data underlying the findings in their manuscript fully available?

Reviewer #1: Yes

Reviewer #2: No

4. Is the manuscript presented in an intelligible fashion and written in standard English?

Reviewer #1: Yes

Reviewer #2: No

5. Review Comments to the Author

Reviewer #1: This manuscript performed a systematic review and meta-analysis to estimate prevalence of respiratory symptoms associated with occupational exposure of dust among Ethiopian factory workers. This study is specifically important as it estimates the overall prevalence of respiratory symptoms among workers in different industries whereas the previous studies were confined to specific industries. However, the authors are requested to address the following comments:

1. Paragraph 6 in the introduction section “The prevalence of occupational…….to 69.8% wood factory workers” has some overlapping/repetitive information with paragraph 2 in the discussion section “According to the findings…………contributing factors to the discrepency.” Either paragraph can be rephrased to avoid it.

2. Paragraph 7 in the introdcution section starts with “However, the research findings are inconsitent.” Which research findings are inconsistent and why do the authors think so?

3. In the last paragraph of introduction section, the authors should focus and elaborate on the scope of this manuscript compared to previous studies i.e. pooled estimates of prevalence in different industries vs. industry specific prevalence in previous studies.

4. Research question should be within the text in the introduction section.

5. It is better to put table 1 in the supplementary information.

6. It is better to discuss the limitations in the discussion section rather than having a separate section.

7. Please revise the manuscript for grammatical/language corrections. For example, there should be no “and” in the first line in the result of the abstract “….studies that met inclusion criteria and were included….” Another example, there is no verb in the last line of fifth paragraph of introdcution section “…The reported rspiratory….magnitude…..to factory/industry”.

Reviewer #2: Dear authors,

I have concern with previous published works, which is more convenient than yours. Also, you need to finalize the statistical analysis in the way all the works presents and graph them accordingly.

6. PLOS authors have the option to publish the peer review history of their article (what does this mean?). If published, this will include your full peer review and any attached files.

Reviewer #1: No

Reviewer #2: No

---

## [Author Response · Author response to Decision Letter 0]

24 Feb 2023

Responses 

Reviewer #1 

1. Paragraph 6 in the introduction section “The prevalence of occupational…….to 69.8% wood factory workers” has some overlapping/repetitive information with paragraph 2 in the discussion section “According to the findings…………contributing factors to the discrepency.” Either paragraph can be rephrased to avoid it. 

Thank for your comment. we have revised our manuscript as per your comment

2. Paragraph 7 in the introdcution section starts with “However, the research findings are inconsitent.” Which research findings are inconsistent and why do the authors think so? 

Thank you a lot. We revised it. It is necessary to pool the prevalence of respiratory symptoms and associated factors at the factory level due to variations in findings across previously existing primary studies.

3. In the last paragraph of introduction section, the authors should focus and elaborate on the scope of this manuscript compared to previous studies i.e. pooled estimates of prevalence in different industries vs. industry specific prevalence in previous studies. 

Thank you very much for your advice. Previous systematic reviews and meta analyses pooled some of the studies conducted outside of industries or outdoor environments, for example, among street sweeping workers. As a result, it is not the correct method because the exposure status and working environment differ in the outdoor and indoor environments. Furthermore, dust control measures differ in the indoor and outdoor environments. Engineering control measures, for example, are inapplicable in an outdoor environment. As a result, estimating pooled prevalence specifically among factory workers was critical in order to implement appropriate prevention and control measures to improve worker and factory owner productivity. Therefore, we conducted this systematic review and meta analysis of studies.

4. Research question should be within the text in the introduction section. 

Thank you. We replaced the research questions with the study's objective.

5. It is better to put table 1 in the supplementary information. 

Thank you for your constructive feedback. Table 1 was added as supplementary information.

6. It is better to discuss the limitations in the discussion section rather than having a separate section. 

Thank you. We revised accordingly.

7. Please revise the manuscript for grammatical/language corrections. For example, there should be no “and” in the first line in the result of the abstract “….studies that met inclusion criteria and were included….” Another example, there is no verb in the last line of fifth paragraph of introdcution section “…The reported rspiratory….magnitude…..to factory/industry”. 

Thank you a lot. We revised the the whole manuscript to correct grammatical and language errors.

Reviewer #2

1. I have concern with previous published works, which is more convenient than yours. Also, you need to finalize the statistical analysis in the way all the works presents and graph them accordingly. 

Thank for your comment. We changed the entire document in the revised version. This study differs from previous ones in that it focuses primarily on factory workers. However, studies conducted among outdoor workers were included in previous systematic reviews and meta analyses. I believe that including studies conducted among street sweeping workers (outdoor environment) was not the correct approach because exposure status, type of dust generated, and control measures implemented were completely different in indoor and outdoor work environments. As a result, combining outdoor and indoor workplace prevalence is not the best approach. We performed various statistical analyses such as meta regression, sensitivity analysis, and Egger's test.

---

## [Decision Letter · Decision Letter 1]

22 Mar 2023

PONE-D-22-30181R1Occupational exposure to dust and Respiratory symptoms among Ethiopian factory workers: a systematic review and meta-analysisPLOS ONE

Dear Dr. Ashuro,

Thank you for submitting your manuscript to PLOS ONE. After careful consideration, we feel that it has merit but does not fully meet PLOS ONE’s publication criteria as it currently stands. Therefore, we invite you to submit a revised version of the manuscript that addresses the points raised during the review process. While the article presents valuable insights, there are a few areas for improvement.

The authors should provide a PRISMA Checklist to enhance the transparency and completeness of their systematic review. 

The article's methodology should include a clear definition of respiratory symptoms. 

The results of the selection process should be reported as usual for this type of study. Specifically, the study's selection process from L170 to 175 should be presented in the results section.

We look forward to receiving your revised manuscript.

Kind regards,

Sebastien Kenmoe

Academic Editor

PLOS ONE

Journal Requirements:

Reviewers' comments:

Reviewer's Responses to Questions

**Comments to the Author**

1. If the authors have adequately addressed your comments raised in a previous round of review and you feel that this manuscript is now acceptable for publication, you may indicate that here to bypass the “Comments to the Author” section, enter your conflict of interest statement in the “Confidential to Editor” section, and submit your "Accept" recommendation.

Reviewer #1: All comments have been addressed

2. Is the manuscript technically sound, and do the data support the conclusions?

Reviewer #1: Yes

3. Has the statistical analysis been performed appropriately and rigorously? 

Reviewer #1: Yes

4. Have the authors made all data underlying the findings in their manuscript fully available?

Reviewer #1: Yes

5. Is the manuscript presented in an intelligible fashion and written in standard English?

Reviewer #1: Yes

6. Review Comments to the Author

Reviewer #1: (No Response)

7. PLOS authors have the option to publish the peer review history of their article (what does this mean?). If published, this will include your full peer review and any attached files.

Reviewer #1: No

---

## [Author Response · Author response to Decision Letter 1]

1 Apr 2023

Point by point response to reviewers

1. The authors should provide a PRISMA Checklist to enhance the transparency and completeness of their systematic review. 

response: Thank for your comment. We provided a PRISMA Checklist as supporting information as per your comment

2. The article's methodology should include a clear definition of respiratory symptoms. 

Response: Thank you very much. In the revised version of the manuscript, we defined respiratory symptoms

3. The results of the selection process should be reported as usual for this type of study. Specifically, the study's selection process from L170 to 175 should be presented in the results section

Response: Thank for your comment. In the revised version, we presented the selection process in the results section.

---

## [Editor Report · Decision Letter 2]

4 Apr 2023

Occupational exposure to dust and Respiratory symptoms among Ethiopian factory workers: a systematic review and meta-analysis

PONE-D-22-30181R2

Dear Dr. Ashuro,

We’re pleased to inform you that your manuscript has been judged scientifically suitable for publication and will be formally accepted for publication once it meets all outstanding technical requirements.

Kind regards,

Sebastien Kenmoe

Academic Editor

PLOS ONE
---

## [Editor Report · Acceptance letter]

13 Apr 2023

PONE-D-22-30181R2 

Occupational exposure to dust and Respiratory symptoms among Ethiopian factory workers: A systematic review and meta-analysis 

Dear Dr. Ashuro:

I'm pleased to inform you that your manuscript has been deemed suitable for publication in PLOS ONE. Congratulations! Your manuscript is now with our production department. 

Kind regards, 

on behalf of

Dr. Sebastien Kenmoe 

Academic Editor

PLOS ONE